# First Records of Picobiine Mites Associated with Birds-of-Paradise: Can Interspecific Sexual Behaviour of Hosts Play a Role in the Distribution of Quill Mite Parasites? [note 1]

**DOI:** 10.3390/ani13091509

**Published:** 2023-04-29

**Authors:** Bozena Sikora, Markus Unsoeld, Roland R. Melzer, Stefan Friedrich, Maciej Skoracki

**Affiliations:** 1Department of Animal Morphology, Faculty of Biology, Adam Mickiewicz University, Uniwersytetu Poznańskiego 6, 61-614 Poznań, Poland; boszka@amu.edu.pl; 2SNSB-Bavarian State Collection of Zoology, Sektion Ornithology, Münchhausenstr. 21, 81247 Munich, Germany; unsoeld@snsb.de; 3SNSB-Bavarian State Collection of Zoology, Sektion Arthropoda Varia, Münchhausenstr. 21, 81247 Munich, Germany; melzer@snsb.de (R.R.M.); friedrich@snsb.de (S.F.); 4Faculty of Biology, Ludwig-Maximilians-Universität München, 82152 Planegg-Martinsried, Germany; 5GeoBio-Center, Ludwig-Maximilians-Universität München, Richard-Wagner-Str. 10, 80333 Munich, Germany

**Keywords:** Acari, birds, birds-of paradise, ectoparasites, quill mites, Picobiinae, Syringophilidae

## Abstract

**Simple Summary:**

We report on the occurrence of parasitic quill mites of the family Syringophilidae on the well-known avian group birds-of-paradise. Our investigation resulted in the discovery of two new species of mites belonging to the subfamily Picobiinae, which has never before been recorded on birds-of-paradise. We hypothesise that the interspecific sexual behaviour of paradisaeids, which involves copulation with non-closely related host species, may have facilitated the spread of syringophilid mites between different and unrelated host species. These findings highlight the potential role of sexual behaviour in the distribution and transmission of parasitic quill mites among avian hosts.

**Abstract:**

While birds-of-paradise (Passeriformes: Paradisaeidae) are a well-known group of birds, our understanding of their parasites is still limited. This study reports on parasitic quill mites of the subfamily Picobiinae (Acariformes: Syringophilidae), which have never before been recorded on this group of birds. The mite specimens presented in this paper were collected from birds-of-paradise that had been captured in Papua New Guinea and Indonesia in the years 1910–1911 and are now deposited in the Bavarian State Collection of Zoology, Munich, Germany. Two syringophilid species are described as new to science: (i) *Picobia frankei* sp. n. from the magnificent riflebird *Lophorina magnifica*, the glossy-mantled manucode *Manucodia ater*, and the crinkle-collared manucode *Manucodia chalybatus*, and (ii) *Gunabopicobia garylarsoni* sp. n. from the twelve-wired bird-of-paradise *Seleucidis melanoleucus* and the lesser bird-of-paradise *Paradisaea minor*. We hypothesise that the presence of both picobiine species on phylogenetically unrelated paradisaeids may be caused by the sexual behaviour of these birds, where interspecific copulations may play a role in the switching of parasites between non-closely related host species.

## 1. Introduction

Birds-of-paradise (Aves: Paradisaeidae) are a passerine family incorporating 44 species in 15 genera distributed in New Guinea, Moluccas, and north-eastern Australia [1,2]. Most of them live in rainforest habitats and are strongly dimorphic and polygynous. Within the superfamily Corvoidea (Corvoid Oscines), the birds-of-paradise are grouped together with the Laniidae (shrikes), Corcoracidae (australian mudnesters), Melampittidae (melampittas), Monarchidae (monarch flycatchers), Corvidae (crows and jays), and the monotypic families Platylophidae (crested jay) and Ifritidae (ifrit), in the clade “crown corvoids” [1,2,3]. Despite birds-of-paradise being a well-known group of birds, our knowledge of their syringophilid parasites is still far from satisfactory, and no research has been conducted on the occurrence of parasitic mites belonging to the subfamily Picobiinae associated with this host family.

Picobiinae Johnston and Kethley, 1973, is one of two subfamilies of highly specialised parasites permanently associated with birds grouped in the family Syringophilidae Lavoipierre, 1953 [4,5]. Unlike representatives of the subfamily Syringophilinae, which mainly inhabit quills of wing and tail feathers, picobiines occupy contour feathers exclusively (except the enigmatic and monotypic genus *Calamincola* Casto, 1977). Picobiines are a taxonomically diverse group of quill mites, with about 80 species grouped in 12 genera described to date [6,7,8]. They are also widely distributed on avian hosts, and to date, picobiines have been recorded from about 200 neognathous bird species belonging to 11 orders and occurring in all zoogeographical regions (except Antarctica) [9]. Considering that only a small portion of extant bird species has been investigated up to now, it is reasonable to assume that the described and known picobiine-fauna is only the tip of the iceberg of the real diversity of this group of mites.

In this paper, we present the results of our studies conducted on mites of the subfamily Picobiinae parasitising birds of the family Paradisaeidae.

## 2. Materials and Methods

The mite material used in this study was collected from dry bird skins of birds-of-paradise captured in Papua New Guinea and Indonesia in the years 1910–1911 and deposited in the ornithological collection of the Bavarian State Collection of Zoology (Munich, Germany). About ten contour feathers (near the cloaca region) were examined under a stereomicroscope from each host specimen. Before mounting, mites were softened in Nesbitt’s solution at room temperature for three days [6]. The mites were then mounted on slides in Hoyer’s medium and examined using a light microscope (ZEISS Axioscope, Oberkochen, Germany) with differential interference contrast (DIC) illumination and a camera lucida drawing attachment.

In the description, all measurements are given in micrometres. The dimension ranges of the paratypes are given in parentheses, following the data from the holotype. The idiosomal setation follows Grandjean [10] as adapted for Prostigmata by Kethley [11]. The leg chaetotaxy follows that proposed by Grandjean [12], and the morphological terminology follows Skoracki [6].

The specimen depositories are cited using the following abbreviations: AMU—A. Mickiewicz University, Department of Animal Morphology, Poznań, Poland; ZSM—Bavarian State Collection for Zoology, Sektion Arthropoda Varia, Munich, Germany.

## 3. Results

### 3.1. Systematics

Family Syringophilidae Lavoipierre

Subfamily Picobiinae Johnston and Kethley

#### 3.1.1. Descriptions

##### *Picobia frankei* sp. n. (Figure 1, Figure 2 and Figure 3)

Female (non-physogastric form) (Figure 1 and Figure 2). Total body length 570 (630 in one paratype). *Gnathosoma*. Hypostomal apex with pair of rounded convexities (Figure 2A). Stylophore 250 (250) long; exposed portion of stylophore apunctate, 155 (160) long. Infracapitulum covered with minute punctations. Each medial branch of peritremes with four or five chambers; each lateral branch with nine chambers (Figure 2B). Movable cheliceral digit 195 (210) long, edentate on proximal end. *Idiosoma*. Propodonotal shield divided into three sclerites: small ovate medial sclerite punctate and situated in middle of propodonotum; two lateral sclerites bearing bases of setae *ve*, *si*, and *se* sparsely punctate near bases of setae *si* (Figure 1A). Setae *vi*, *ve*, and *si* are distinctly ornamented and knobbed. Hysteronotal shield absent. Hysteronotal setae subequal in length. Pygidial shield is large, entire and apunctate. Genital plate apunctate. Genital setae short and stout (Figure 2E). Genital lobes absent. Pseudanal setae *ps1* 1.4 times longer than *ps2*. Setae *ag2* situated postero-lateral to *ag1*. Coxal fields I-IV well sclerotised and apunctate. Cuticular striations as in Figure 1A, B. *Legs*. Setae *tc”* of tarsi III and IV slightly longer (1.3 times) than *tc’III-IV* (Figure 2C). Antaxial and paraxial members of claw pair III–IV equal in size and shape (Figure 2D). Solenidia of legs I as in Figure 2F.

Lengths of setae: *vi* 210 (200), *ve* (210), *si* 200 (220), *se* 320 (310), *c1* 300 (280), *c2* 310 (295), *d1* 175 (165), *d2* 280 (290), *e2* 155, *f1* 45 (40), *f2* 65 (60), *h1* 70 (75), *h2* (545), *ag1* 115, *ag2* 65, *ag3* 175, *g1* 15 (20), *ps1* 35 (40), *ps2* 25 (30), *3b* (40), *3c* (95), *4c* 145, *lRIV* 40, *tc’III-IV* (70), *tc”III-IV* (90).

Male (Figure 3). Total body length 580–590 in six paratypes. *Gnathosoma*. Hypostomal apex smooth (Figure 3C). Stylophore 120–125 long. Infracapitulum apunctate. Each medial branch of peritremes with three or four chambers, and each lateral branch with nine or ten chambers (Figure 3D). Movable cheliceral digit 85–95 long, edentate on proximal end. *Idiosoma*. Propodonotal shield divided into three sclerites: small, ovate medial sclerite apunctate, situated in middle of propodonotum; two lateral sclerites bearing bases of setae *ve*, *si*, and *se* punctate in anterior part (Figure 3A). Hysteronotal shield entire, punctate near bases of setae *e2*, with deep cleft on anterior margin, bearing bases of setae *d1* and *e2*. Setae *d2* approx. ten times longer than *d1* and *e2*. Pygidial shield entire, apunctate and with rounded anterior margin, bearing bases of setae *f2* and *h2*. Genital and pseudanal setae surrounding genital opening as in Figure 3E. Coxal fields I–IV well sclerotised, apunctate. Bases of setae *ag1* situated on the posterior margin of large, oval, punctate aggenital plates (Figure 3B). Setae *ag1* approximately twice as long as *ag2*. Cuticular striations as in Figure 3A,B. *Legs*. Solenidia of legs I as in Figure 3F.

Lengths of setae: *vi* 125–155, *ve* 115–130, *si* 160–180, *se* 220–235, *c1* 230–245, *c2* 205–230, *d1* 20–30, *d2* 210–215, *e2* 20–30, *f2* 15–20, *h2* 240–295, *ag1* 80–85, *ag2* 30–35, *3b* 40, *3c* 50–65, *4b* 40–45, *4c* 85–90, *lRIII* 35–40, *lRIV* 25–35.

##### Hosts and Distribution

Birds of the family Paradisaeidae: magnificent riflebird *Lophorina magnifica* (Vieillot), glossy-mantled manucode *Manucodia ater* (Lesson) from Papua New Guinea, and crinkle-collared manucode *Manucodia chalybatus* (Pennant) from Indonesia and Papua New Guinea (Figure 4).

##### Type Material

Female holotype and paratypes: one female (non-physogastric form), one female (physogastric form), and six males collected from quills of contour feathers of the magnificent riflebird *Lophorina magnifica* (Vieillot) (host reg. no. ZSM-13.512, female); PAPUA NEW GUINEA: Western Highlands Province, Bismarck Range, coll. Blum.

##### Type Material Deposition

Holotype and paratypes are deposited in the SNSB-ZSM (reg. no. ZSMA20230001), except one female paratype (physogastric form) and one male paratype in the AMU (reg. no. AMU MS 21-0907-071).

##### Additional Material

Ex the glossy-mantled manucode *Manucodia ater* (Lesson) (host reg. no. ZSM-11.580, male); PAPUA NEW GUINEA: June 1910, coll. L. von Wiedenfeld: two females (physogastric form) deposited in the SNSB-ZSM (reg. no. ZSMA20230002) and two females (physogastric form) (reg. no. AMU MS 21-0907-076).

Ex the crinkle-collared manucode *Manucodia chalybatus* (Pennant) (host reg. no. ZSM-14.735, female); INDONESIA: Raja Ampat archipelago, Misool Isl., October 1911, coll. O. Tauern: one female (non-physogastric form), one female (physogastric form) and five males deposited in the AMU (reg. no. AMU MS 21-0907-078).

From same host species (host reg. no. ZSM-11.582, female); PAPUA NEW GUINEA: February 1910, coll. L. von Wiedenfeld: one female (non-physogastric form), four females (physogastric form), and one male deposited in the SNSB-ZSM (reg. no. ZSMA20230003), five females (physogastric form) and one male in the AMU (reg. no. AMU MS 21-0907-079). From the same host species (host reg. no. ZSM-11.585, male); PAPUA NEW GUINEA: 3 March 1910, coll. L. von Wiedenfeld: one female (non-physogastric form), one female (physogastric form), and four males deposited in the SNSB-ZSM (reg. no. ZSMA20230004), two females (physogastric form) and three males in the AMU (reg. no. AMU.SYR.893).

##### Differential Diagnosis

*Picobia frankei* sp. n. is morphologically similar to *P. lamprotornis* Klimovicova, Skoracki, Wamiti, and Hromada, 2014, described from the superb starling *Lamprotornis superbus* Rueppell (Passeriformes: Sturnidae) [13]. In females of both species, the hypostomal apex is with a pair of rounded convexities; the propodonotal shield is divided into three sclerites; the pygidial shield and genital plate are well-developed; genital setae are short and stout; aggenital setae *ag2* are situated postero-lateral to *ag1*, and the coxal fields are apunctate. The new species differs from *P. lamprotornis* by the following features: in females of *P. frankei*, setae *ps1* are distinctly longer than *ps2*; the lengths of setae *se*, *c2*, and *ag1* are 310–320, 295–310, and 115, respectively; antaxial and paraxial members of claw pair III–IV are equal in size; in males, the infracapitulum is apunctate; each lateral branch of the peritremes has nine or ten chambers; the lengths of setae *d2* and *ag1* are 210–215 and 80–85, respectively. In females of *P. lamprotornis*, setae *ps1* and *ps2* are subequal in length; the lengths of setae *se*, *c2*, and *ag1* are 190, 175–180, and 40–45, respectively; antaxial and paraxial members of claw pair III–IV are unequal in size; in males, the infracapitulum is punctate; each lateral branch of the peritremes has seven or eight chambers; the lengths of setae *d2* and *ag1* are 130 and 35–45, respectively.

##### Etymology

This species is named in honour of the German botanist, the great naturalist and our friend Dr Thassilo Franke (SNSB, Munich, Germany).

##### *Gunabopicobia garylarsoni* sp. n. (Figure 5 and Figure 6)

Female (Figure 5). Total body length 730 (725–750 in three paratypes). *Gnathosoma*. Hypostomal apex rounded, two small and sharp-ended protuberances present (Figure 5D). Stylophore 230 (225–240) long; exposed portion of stylophore apunctate, 140 (140–150) long. Infracapitulum apunctate. Each medial branch of peritremes with six chambers, and each lateral branch with eight or nine chambers (Figure 5C). Movable cheliceral digit 180 (180–195) long. *Idiosoma*. Propodonotal shield divided into three sclerites: medial sclerite with anterior end extending to midlength between setae *ve* and *si*, posterior margin bearing bases of setae *c1*; two lateral sclerites wide, punctate and with patches, bearing bases of setae *vi*, *ve*, *si*, and *se* (Figure 5A). Setae *vi*, *ve*, and *si* slightly ornamented (Figure 5F). Hysteronotal shield absent. Hysteronotal setae subequal in length. Pygidial shield reduced to small and apunctate sclerite connecting bases of setae *f1*. Setae *f1* and *h2* long, *f2* and *h1* short. Bases of setae *1a*–*1a* in close proximity, but not coalesced. Coxal fields I–IV well sclerotised and punctate. Aggenital and genital plates absent. Bases of setae *ag1* situated anterior to *ag2*. Pseudanal setae *ps1* and *ps2* subequal in length. Cuticular striations as in Figure 5A,B. *Legs*. Antaxial and paraxial members claw pair III–IV unequal in size (Figure 5G). Solenidia of legs I as in Figure 5F.

Lengths of setae: *vi* 25 (25), *ve* 95 (95–115), *si* 130, *se* 235 (230–280), *c2* (240–280), *d1* 240 (240–255), *d2* 235 (230–275), *e2* 220 (235–250), *f1* longer than 300, *f2* 35 (35–40), *h1* 30 (25–30), *h2* (400–435), *ag1* 185 (180–200), *ag2* 50 (50–55), *ag3* 200 (170–200), *ps1* and *ps1* 10 (10–15), *tc’III–IV* 80 (70–80), *tc”III–IV* 80 (70–80), *3b* 20 (20–30), *3c* 100 (100–120), *4b* (30–40), *4c* (110–145), *lRIII* 20 (20–30), *lRIV* (20–30).

Male (Figure 6). Total body length 530–545 in seven paratypes. *Gnathosoma*. Stylophore 125–135 long; exposed portion of stylophore apunctate, 100–105 long. Infracapitulum apunctate. Movable cheliceral digit 85 long. *Idiosoma*. Propodonotal shield divided into three sclerites: medial sclerite with anterior end reaching midlength between levels of setae *ve* and *si*, posterior margin bearing bases of setae *c1*; two lateral sclerites wide, punctate and with patches, bearing bases of setae *vi*, *ve*, *si*, and *se*. Hysteronotal shield entire, punctate, and with concave anterior margin, bearing bases of setae *d1* and *e2*. Setae *d2* approximately ten times longer than *d1* and *e2*. Pygidial shield entire, punctate and with rounded anterior margin, bearing bases of setae *f2* and *h2*. Coxal fields I–IV well-sclerotised, I–III punctate, IV sparsely punctate or apunctate. Two small, punctate, and egg-shaped aggenital shields situated anterior to level of setal bases *ag1*. Setae *ag1* approximately three times longer than *ag2*. Cuticular striations as in Figure 6.

Lengths of setae: *vi* 40–45, *ve* 70–75, *si* 95–100, *se* 180–185, *c2* 170–185, *d1* 15, *d2* 180–200, *e2* 10–15, *f2* 10–15, *h2* longer than 200, *ag1* 105–115, *ag2* 25–30, *3b* 20, *3c* 55–65, *4b* 20, *4c* 75–80.

##### Hosts and Distribution

Birds of the family Paradisaeidae: twelve-wired bird-of-paradise *Seleucidis melanoleucus* (Daudin) and lesser bird-of-paradise *Paradisaea minor* Shaw, both from Papua New Guinea (Figure 7).

##### Type Material

Female holotype and paratypes: three females and five males collected from quills of contour feathers of the twelve-wired bird-of-paradise *Seleucidis melanoleucus* (Daudin) (host reg. no. ZSM-11.598, female); PAPUA NEW GUINEA: August 1910, coll. L. von Wiedenfeld. Two male paratypes from the same host species (host reg. no. ZSM-11.597, female) and locality. One female paratype from the same host species (host reg. no. ZSM-uncatalogued) and locality; no other data.

##### Type Material Deposition

Holotype and paratypes are deposited in the SNSB-ZSM (reg. no. ZSMA20230005), except two female and two male paratypes in the AMU (reg. no. AMU MS 21-0907-068, 069, 070).

##### Additional Material

Ex the lesser bird-of-paradise *Paradisaea minor* Shaw (host reg. no. SNSB-ZSM 11.589, male); PAPUA NEW GUINEA: July 1910, coll. L. von Wiedenfeld: two females and four males are deposited in the SNSB-ZSM (reg. no. ZSMA20230006), and three females and three males in the AMU (reg. no. AMU MS 21-0907-081).

##### Differential Diagnosis

This new species is morphologically similar to *G. metropelia* Kaszewska et al., 2018 described based on females and collected from the black-winged ground-dove *Metriopelia melanoptera* (Molina) (Columbiformes: Columbidae) in Argentina [14] by the presence of long setae *f1* and not coalesced bases of setae *1a*–*1a* (setae *f1* are short and bases of setae *1a*–*1a* are coalesced in all other species), and differs by the following features: in females of *G. garylarsoni*, the propodonotal shield is divided into three sclerites (two lateral and one medial); the lengths of setae *vi*, *ve*, and *si* are 25, 95–115, and 130, respectively; aggenital setae *ag2* are shorter than *ag1*, and the lengths of *ag1* and *ag2* are 180–200 and 50–55, respectively. In females of *G. metropelia*, the propodonotal shield is entire; setae *vi*, *ve*, and *si* are short (about 10–20) and subequal in length; aggenital setae *ag2* are longer than *ag1*, and the lengths of *ag1* and *ag2* are 40–55 and 80, respectively.

##### Etymology

This species is named in honour of famous cartoonist Gary Larson, whose cartoons are loved by biologists around the globe and who once bemoaned that a mallophagan [15] and not a new swan species was named in honour of him. Now it’s at least a parasite of a bird-of-paradise.

## 4. Discussion

Case of *Picobia frankei.* The genus *Picobia* belongs to the *Picobia*-generic-group, including five genera, and represents the most species-rich genus in the family Picobiinae with 44 known species [8]. Most representatives of this genus are associated with birds of the order Passeriformes (40 species), while only a small part of the *Picobia* fauna has been recorded from non-passeriform birds, such as Piciformes (three species), and Bucerotiformes (one species) [7,8]. Given numerous associations of the genus *Picobia* with passerine hosts, our finding of a new species from this genus on birds-of-paradise is not unexpected. On the other hand, considering that most picobiine species are recorded from one or closely related hosts (e.g., belonging to the same host genus) [6,7], it is interesting that *Picobia frankei* inhabits host species from different and phylogenetically not closely related lineages of Paradisaeidae [16]. In our studies, this mite was found on manucodes (*Manucodia ater* and *M. chalybatus*), which belong to a basal lineage of paradisaeids, and on riflebird (*Lophorina magnifica*), a distantly more derived host species (see Figure 8). We hypothesise that the distribution of *P. frankei* among these host species may be associated with the behaviour of its hosts. Some birds-of-paradise species mix in flocks with other species of Paradiesaeidae and birds of other families in search of food and to avoid predator attacks [17]. These mixed-species flocks could provide opportunities for hybridisations between representatives of different species and even genera of Paradiesaeidae [17,18,19,20,21,22,23,24]. Such interspecific copulations could play a role in the switching of parasites between distantly related host species.

Case of *Gunabopicobia garylarsoni.* The discovery of a representative of the genus *Gunabopicobia* on birds-of-paradise is both unexpected and puzzling. Until now, seven species of this genus have been recorded exclusively on pigeons and doves (order Columbiformes) [7,14,25,26], which are phylogenetically distant from passerines [27,28]. Among them, two species, *G. masalaje* Kaszewska et al. (2014) and *G. lathami* Kaszewska et al. (2014) have been recorded on columbiform birds in Papua New Guinea (e.g., the nicobar pigeon *Caloenas nicobarica* (Linnaeus), the purple-tailed imperial-pigeon *Ducula rufigaster* (Quoy and Gaimard), the island imperial-pigeon *D. pistrinaria* bonaparte, the white imperial-pigeon *D. luctuosa* (Temminck), or the orange-bellied fruit-dove *Ptilinopus iozonus* gray) [14]. The occurrence of this quill mite species on four infested paradisaeid birds (three specimens of *Seleucidis melanoleucus* and one specimen of *Paradisaea minor*) cannot be attributed to accidental museum contamination but rather to host-switching of its ancestor from columbiform birds. Notably, this mite species, similar to the aforementioned one, also inhabits not closely related paradisaeid taxa, representatives of the genera *Paradisaea* and *Seleucidis* (Figure 8). This fact suggests that the occupation of hosts belonging to phylogenetically distant clades could also be caused by their sexual behaviour. In fact, the “Wonderful Bird of Paradise” [18] is thought to be a hybrid of *Paradisaea minor* and *Seleucidis melanoleuca*, which may provide further evidence for this hypothesis. Particularly, infested “dominant” males of polygynous species may act as effective “spreaders” of syringophilids due to their frequent contact with many females of their own species and others.

## 5. Conclusions

Mites of the family Syringophilidae have a strong association with their avian hosts, and the majority of these mites are mono- or oligoxenous parasites that are expressed in restriction of mite species to a particular host species or a closely related group of hosts in terms of their phylogenetic origin [5,6,7]. These mites are obligate and permanent parasites whose biology is strongly linked to their avian hosts. Therefore, their diversification pattern often mirrors that of their hosts. However, with the increasing number of studies on syringophilid mites, there are more and more examples of horizontal transfers where quill mite parasites jump between unrelated host groups. The flagship examples can be mites switching between host-predator and host-prey [29] or during the sharing of the same ecological niche by different host species, especially of gregarious birds [30]. Furthermore, the suggestion that horizontal transmission of mite parasites during bird copulation may play a role has been presented for quill mites infesting brood parasitic cuckoos [31]. In our studies, except the description of two new species of picobiine mites, we hypothesise that the interspecific sexual behaviour of paradisaeids, which involves copulation with non-closely related host species, may have facilitated the spread of syringophilid mites between different and unrelated host species. These findings highlight the potential role of sexual behaviour in the distribution and transmission of parasitic quill mites among not related avian hosts.

## Figures and Tables

**Figure 1 animals-13-01509-f001:**
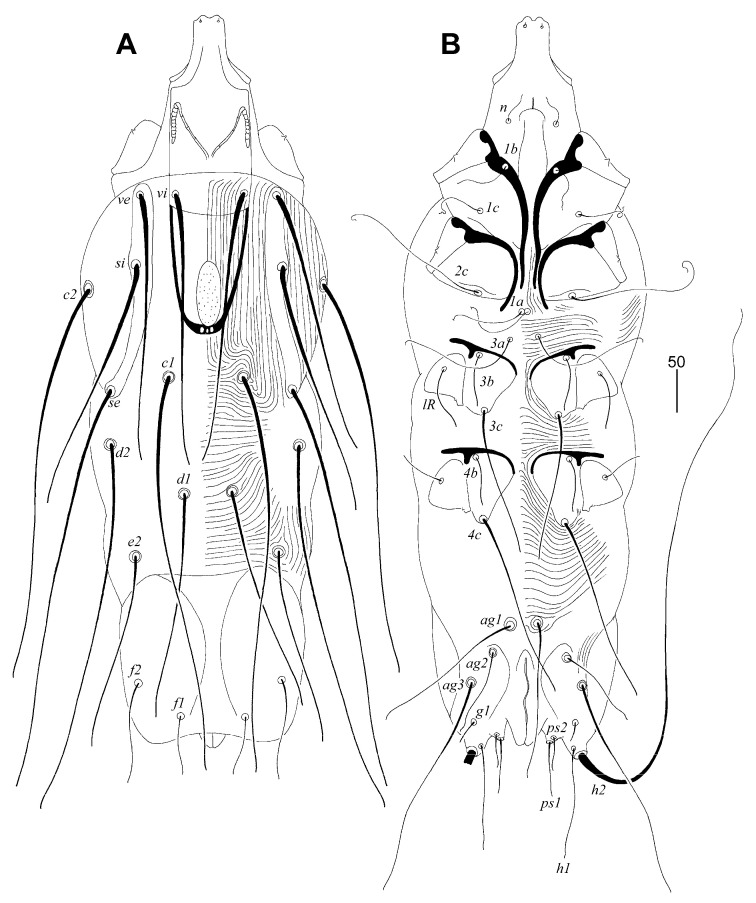
*Picobia frankei* sp. n., female. (**A**)—dorsal view; (**B**)—ventral view.

**Figure 2 animals-13-01509-f002:**
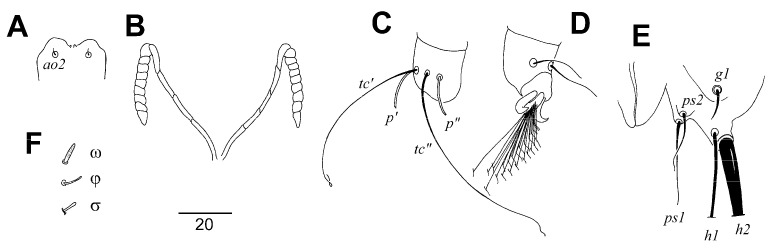
*Picobia frankei* sp. n., female. (**A**)—hypostomal apex; (**B**)—peritremes; (**C**)—tarsus III in dorsal view; (**D**)—tarsus III in ventral view; (**E**)—opisthosoma in ventral view; (**F**)—solenidia of legs I.

**Figure 3 animals-13-01509-f003:**
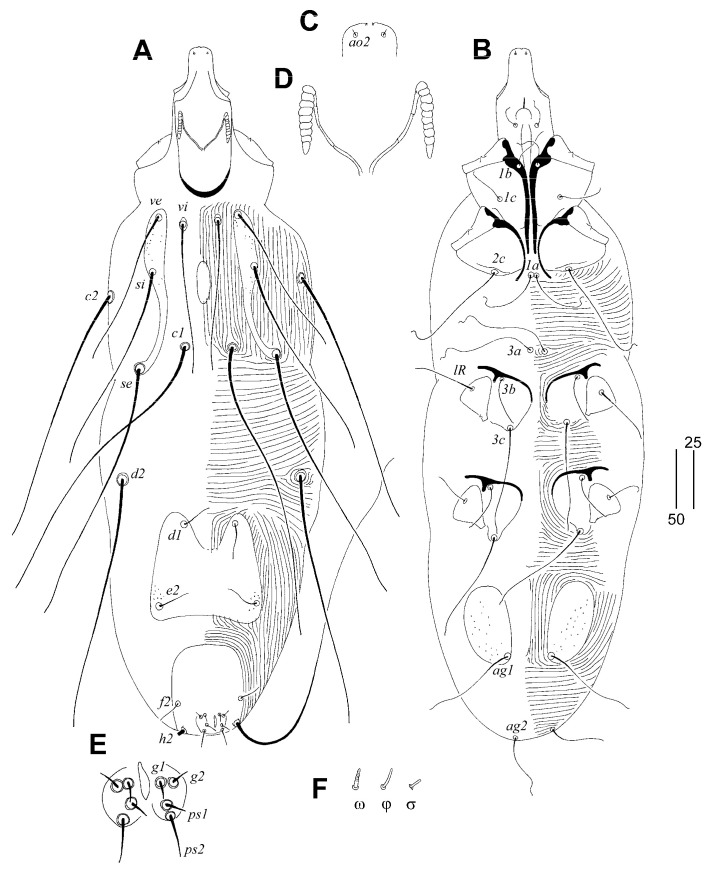
*Picobia frankei* sp. n., male. (**A**)—dorsal view; (**B**)—ventral view; (**C**)—hypostomal apex; (**D**)—peritremes; (**E**)—genital opening; (**F**)—solenidia of legs I. Scale bars: (**A**,**B**) = 50 µm, (**C**–**F**) = 25 µm.

**Figure 4 animals-13-01509-f004:**
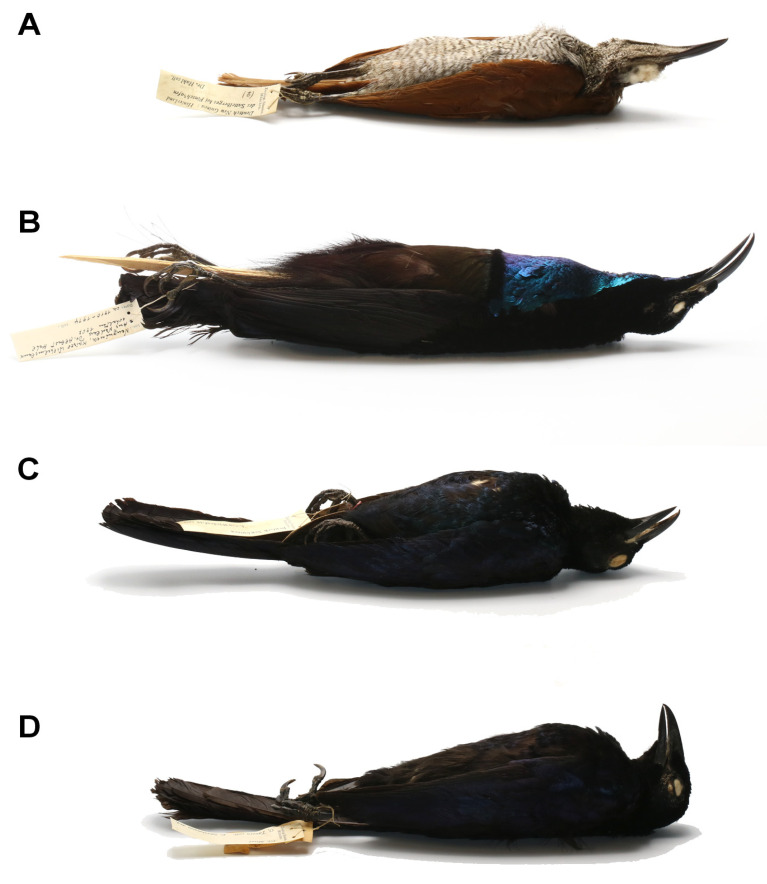
Species of birds-of-paradise infested by *Picobia frankei* sp. n. (**A**)—*Lophorina magnifica* (Vieillot), female; (**B**)—*Lophorina magnifica* (Vieillot), male; (**C**)—*Manucodia ater* (Lesson), male; (**D**)—*Manucodia chalybatus* (Pennant), male.

**Figure 5 animals-13-01509-f005:**
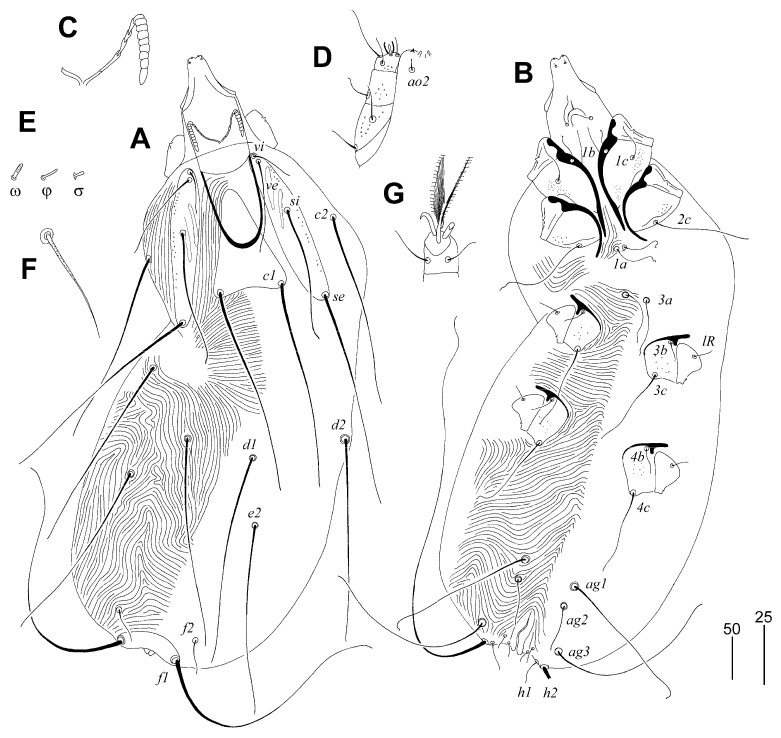
*Gunabopicobia garylarsoni* sp. n., female. (**A**)—dorsal view; (**B**)—ventral view; (**C**)—peritreme; (**D**)—gnathosoma in ventral view; (**E**)—solenidia of legs I; (**F**)—seta *vi*; (**G**)—tarsus III in ventral view. Scale bars: (**A**,**B**) = 50 µm, (**C**–**G**) = 25 µm.

**Figure 6 animals-13-01509-f006:**
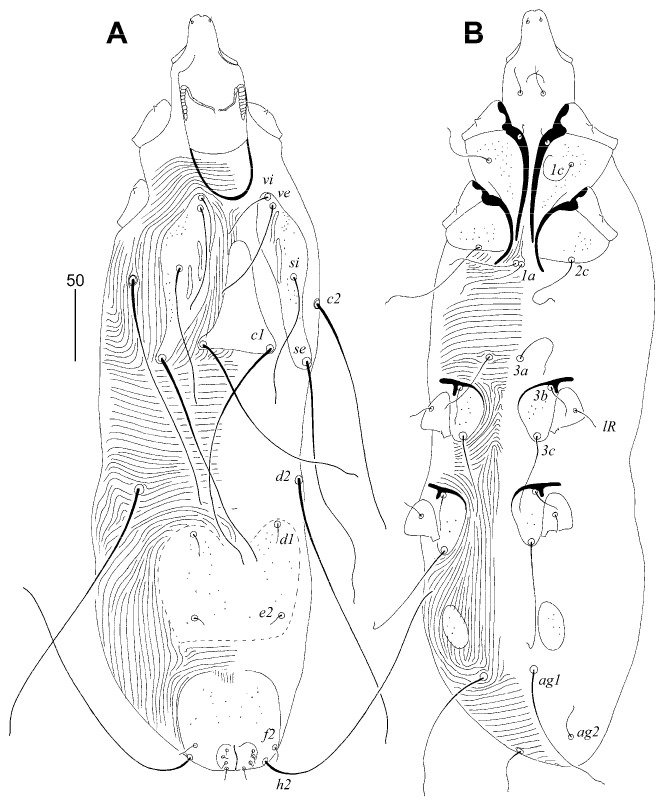
*Gunabopicobia garylarsoni* sp. n., male. (**A**)—dorsal view; (**B**)—ventral view.

**Figure 7 animals-13-01509-f007:**
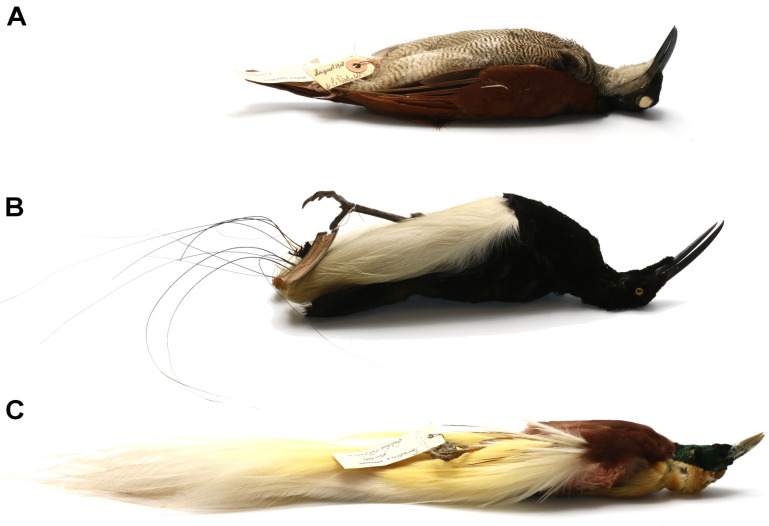
Species of birds-of-paradise infested by *Gunabopicobia garylarsoni* sp. n. (**A**)—*Seleucidis melanoleucus* (Daudin), female; (**B**)—*Seleucidis melanoleucus*, male (Daudin); (**C**)—*Paradisaea minor*, male Shaw.

**Figure 8 animals-13-01509-f008:**
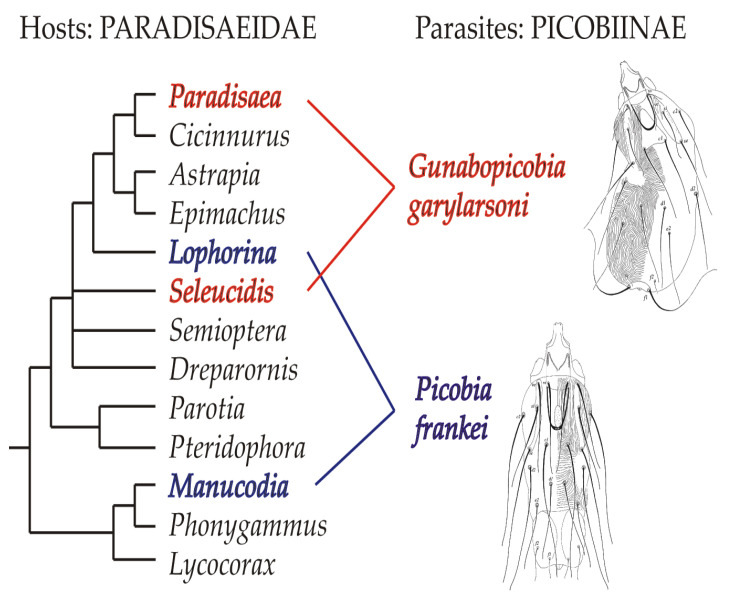
Distribution of picobiine mites on birds-of-paradise (phylogenetic relationships among genera after Irestedt et al. [14], modified).

## Data Availability

Data is available upon request from the corresponding author.

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
