# Peer review of "First Records of Picobiine Mites Associated with Birds-of-Paradise: Can Interspecific Sexual Behaviour of Hosts Play a Role in the Distribution of Quill Mite Parasites?†"

_animals, 2023, doi:10.3390/ani13091509_

Round 1

Reviewer 1 Report

The reviewed manuscript presents descriptions of two new syringophilid mites of the subfamily Picobiinae recorded for the first time from birds-of-paradise and draws out an interesting and quite probable hypothesis explaining the host associations of both new species with non-sister genera of paradisaeids. The manuscript deserves to be published after a minor revision. In general, it mostly needs just re-editing of some phrases in descriptions and discussion. The most important comments are provided below; other notes and corrections are given in the attached text of the manuscript.

 P.2, lines 48-50. “Despite being a well-known group of birds, our understanding of their mite parasites is still far from satisfactory, and no research has been conducted on the  occurrence of parasitic mites belonging to the subfamily Picobiinae associated with this host family.”

 Ref: It is not clear, what the authors actually mean in some words of this phrase. Do the words “mite parasites” mean all groups of parasitic Acari associated with these hosts (various Prostigmata and Astigmata, ticks, chiggers etc.), or only mites of the family Syringophilidae?

Based on the fragment “ … research has been conducted on the occurrence of parasitic mites belonging to the subfamily Picobiinae”,  it is possible to conclude that investigations of syringophilines have been conducted, while only picobiine mites have never been searched in these birds.

P. 2, line 56. “Picobiines are a taxonomically diverse group of quill mites, with about 80 species grouped in 12 genera described to date.”

Ref: However, the subfamily Syringophilinae is much more diverse with over 160 species in about 40 genera (e.g. see Glowska et al. 2015 – World checklist of quill mites).

P. 3, lines 98, 99. “Propodonotal shield divided into three sclerites - small, oval me dial sclerite punctate, situated in middle of propodonotum; two lateral sclerites sparsely punctate near bases of setae si, bear bases of setae ve, si, and se.”

Ref: It would be reasonable to re-edit this phrase as follows: Propodonotal shield divided into three sclerites: small oval-shaped medial sclerite punctate and situated in middle of propodonotum; two lateral sclerites bearing bases of setae ve, si, and se and sparsely punctate near bases of setae si.

P. 3, line 106.“Antaxial and paraxial claw pairs equal in size and shape (Fig. 2D).”

Ref: From this context, it is possible to uimagine that there exist antaxial and paraxial PAIRS of claws. Apparently, the authors mean that the antaxial and paraxial members in the  claw pairs on tarsal apices are similar. The phrase should be edited.

P. 6, line 198. “Hypostomal apex rounded, two small and sharp-ended protuberances present (Fig. 5D).”

 Ref: In Fig. 5D, it is visible that the apex is not rounded, but has a pair of short rounded convexities.

P. 10, lines 304-308. “Some Birds-of-paradise species mix in flocks with other species of Paradiesaeidae and other families in the search for food and to avoid predator attacks [20]. These mixed-species flocks could provide opportunities for hybridisations between representatives of different species and even genera of Paradiesaeidae [21]. Such interspecific copulations could play a role in the switching of parasites between distantly related host species.” 

Ref: Does the second reference [21] (Fuller 1995) says only about some theoretical “opportunity of hybridization” or directly report about really observed facts of interspecific copulation or another physical contacts between birds-of-paradise of different species in mixed flocs? It is necessary to say that exactly in the discussion. Regarding the transfer of syringophilids, it is even not important, whether the “contact” between different species of paradisaeids has been resulted by the appearance of hybrids. Only the simple physical contact of plumages of bird individuals, at least for a very short time, is only important for successful interspecific (horizontal in evolutionary sense) transfer of syringophilids between different host species.

P. 10, Lines 321, 322. “The occurrence of this quill mite species on four infested paradisaeid birds (three specimens of Seleucidis melanoleucus and one specimen of Paradisaea minor) cannot be attributed to accidental museum contamination but rather to host-switching from columbiform birds.”

Ref: It would be reasonable to write more exactly - “host switching of its ancestor from columbiform …”. Otherwise, it is necessary to admit that “this quill mite species” also presently occurs on some columbids, unknown to us, but living in the same region. Since this mite is already distributed on unrelated paradisaeid hosts, this can mean that host switching from columbids took place relatively long ago. 

The qulity of English is good, just some phrases need some corrections, - see the attached text file. 

Author Response

Dear Editor and dear Reviewer

We want to thank the reviewer for all the valuable comments. Below we give our answers to the comments and remarks:

Dear Editor and Dear Reviewer,

below we give our answers to the comments and suggestions of Reviewer 1:

_______

Reviewer: ...it is not clear, what the authors actually mean in some words of this phrase. Do the words “mite parasites” mean all groups of parasitic Acari associated with these hosts (various Prostigmata and Astigmata, ticks, chiggers etc.), or only mites of the family Syringophilidae?

Answer: We agree, and it should be: ...of their syringophilid parasites is still far from satisfactory... [it is corrected].

Reviewer. ... However, the subfamily Syringophilinae is much more diverse with over 160 species in about 40 genera

Answer: We did not write that Picobiinae is the most diverse, only that they are diverse. [the text has been left unchanged]

Reviewer. ... It would be reasonable to re-edit this phrase as follows: Propodonotal shield divided into three sclerites: small oval-shaped medial sclerite punctate and situated in middle of propodonotum; two lateral sclerites bearing bases of setae vesi, and se and sparsely punctate near bases of setae si.

Answer: We agree, and it was corrected.

Reviewer. ... From this context, it is possible to uimagine that there exist antaxial and paraxial PAIRS of claws. Apparently, the authors mean that the antaxial and paraxial members in the  claw pairs on tarsal apices are similar. The phrase should be edited.

Answer: We agree, and it was corrected.

Reviewer. ...  In Fig. 5D, it is visible that the apex is not rounded, but has a pair of short rounded convexities.

Answer: We don’t agree; the hypostomal apex is rounded and with two sharp-ended protuberances [the text has been left unchanged].

Reviewer. ... Does the second reference [21] (Fuller 1995) says only about some theoretical “opportunity of hybridization” or directly report about really observed facts of interspecific copulation or another physical contacts between birds-of-paradise of different species in mixed flocs? …

Answer: Thank you for this remark. We added more publications which support our hypothesis.

 Reviewer. It would be reasonable to write more exactly - “host switching of its ancestor from columbiform…”. Otherwise, it is necessary to admit that “this quill mite species” also presently occurs on some columbids, unknown to us, but living in the same region. Since this mite is already distributed on unrelated paradisaeid hosts, this can mean that host switching from columbids took place relatively long ago.

Answer: Thank you for this comment. We changed it.

Reviewer 2 Report

The authors describe two new species of quill mites (Syringophilidae), which represent the first members of the subfamily Picobiinae reported from birds of the passeriform family Paradiseidae. The illustrations and verbal descriptions are of very high quality, and I have only a few minor suggestions for modifications or clarifications (see comments on attached manuscript file). In addition to describing the two new species, the authors comment on the unsual host distributions of the mites: both were found on museum skins of distantly related paradiseid hosts, and one comes from a genus that has previously only been recorded from birds from a different order, the Columbiformes. The authors hypothesize that the mites were transferred among distantly related Bird-of-paradise hosts through attempted interspecific copulation. They arebstumped as to history of the apparent broad host-jump from Columbidae to Paradiseadae, but discounted the possiblility that it was due to contamination of paradiseids by columbid mites in museum drawers (presumably because the museum they worked in sensibly doesn't store birds of different orders in the same drawers). I have an alternative suggestion for the source of apparent broad host-use by these mite species. At the time that the birds were collected more than a century ago, there may have been little attempt to separate the birds after they were shot, but rather they may have been stored together in bags. If the mites of these hosts were still alive, they may have transferred among hosts during the time they were in the bag. This may have allowed transfers both among members of the Paradiseidae and between collected columbiforms and paradiseids. This is at least as plausible as interspecific copulations among paradiseids, given the highly species-specific coursthip behaviour in this group, and the choosiness of Bird-of-paradise females. The authors should seek out stronger literature support of interspecific copulation attempts among Birds-of-paradise species to give their hypothesis more credence. 

Other than that, I noted that Figures 4 and 6, and Reference 19, appear not to be cited in the text. Please see annotations on the uploaded file for these and other minor comments and suggestions.

See comments in attached file.

Author Response

Dear Editor and dear Reviewer,

We thank Reviewer_2 for his valuable comments and suggestions. Below we give our answers:

Reviewer. I have an alternative suggestion for the source of apparent broad host-use by these mite species. At the time that the birds were collected more than a century ago, there may have been little attempt to separate the birds ...

Answer: This alternative suggestion can be used for feather mites or mallophagans but, in our opinion, not for quill mites. In our samples, we have found well-developed infapopulations of these parasites in the quills. It takes time for the mites to enter the feather and build up an infrapopulation in the quills (in living hosts).

Reviewer. The authors should seek out stronger literature support of interspecific copulation attempts among Birds-of-paradise species to give their hypothesis more credence.

Answer: Thank you for this suggestion. We added more papers about the hybridization of birds of paradise.